# Latent Causal Probing:
# A Formal Perspective on Probing with Causal Models of Data

**Charles Jin**[*]
MIT CSAIL
ccj@csail.mit.edu

## Abstract

As language models (LMs) deliver increasing performance on a range of NLP tasks, *probing classifiers* have become an indispensable technique in the effort to better understand their inner workings. A typical setup involves (1) defining an auxiliary task consisting of a dataset of text annotated with labels, then (2) supervising small classifiers to predict the labels from the representations of a pretrained LM as it processes the dataset. A high probing accuracy is interpreted as evidence that the LM has learned to perform the auxiliary task as an unsupervised byproduct of its original pretraining objective. Despite the widespread usage of probes, however, the robust design and analysis of probing experiments remains a challenge. We develop a formal perspective on probing using *structural causal models* (SCM). Specifically, given an SCM which explains the distribution of tokens observed during training, we frame the central hypothesis as whether the LM has learned to represent the latent variables of the SCM. Empirically, we extend a recent study of LMs in the context of a synthetic grid-world navigation task, where having an exact model of the underlying causal structure allows us to draw strong inferences from the result of probing experiments. Our techniques provide robust empirical evidence for the ability of LMs to induce the latent concepts underlying text.

## 1 Introduction

As large LMs pretrained on massive amounts of unlabeled text continue to reach new heights in NLP tasks (and beyond), the question of what kinds of information such models encode about their training data remains a topic of intense discussion and research. One prominent technique is to supervise small *probing classifiers* to extract some linguistically relevant property from the representations of the pretrained LM (Shi et al., 2016; Adi et al., 2017; Alain & Bengio, 2018), with the intuition being that the success of the probe reveals the LM has, in fact, learned to encode the property of interest as a byproduct of its training.

Despite their widespread usage, however, probes themselves are also an active area of research, with a number of interconnected open questions in the design and interpretation of probing experiments (Belinkov, 2022), including:

**(Q1) Control and interpretation.** Given that the probe itself is directly supervised to perform the auxiliary task, the observed outcomes could depend not only on the information inherently encoded in the LM but also the ability of the probe to extract the information itself. For instance, researchers have found that training probes to predict *randomized* labels can often yield comparably high accuracies on certain tasks, calling into question the significance of prior results (Hewitt & Liang, 2019). As a result, drawing robust conclusions from the classification accuracy of a probe remains up for debate.

**(Q2) Classifier selection and training.** To combat the risk of measuring the probe's capacity to learn the auxiliary task, researchers often limit probes to low capacity architectures such

---

[*]We gratefully acknowledge the contributions of senior author Martin Rinard, who was unfortunately left off the official author list due to a late-night clerical error.

as linear classifiers (Maudslay et al., 2020). However, other works have countered with evidence that LMs encode more complex concepts using non-linear representations, which can only be accurately measured using higher capacity classifiers (Belinkov & Glass, 2019; Li et al., 2022). A related question which has received little attention is how the training procedure itself (e.g., optimizer selection, training hyperparameters, auxiliary dataset size) interacts with the outcome of the probing experiment.

**(Q3) Auxiliary task design.** Finally, as large, pretrained LMs have progressed from producing human-like text to exhibiting increasingly "intelligent" behaviors such as reasoning and in-context learning (Brown et al., 2020), there is an emerging need to better understand the limitations and capabilities of LMs along dimensions such world knowledge and theory of mind. These domains present a distinct set of challenges compared to traditional linguistic tasks such as part-of-speech tagging and dependency parsing.

The theoretical section of this paper develops a formal perspective on probing using the language of *structural causal models* (SCM). Specifically, given a causal model which explains the distribution of tokens observed during training, we pose the central hypothesis as determining whether the LM has learned to represent the *latent variables* of the SCM: concepts that explain how the text was generated, but are never directly observed during training. We then introduce probes as a means of empirically testing such hypotheses, by extracting the value of the latent concepts given only the LM representations as input. Our setting naturally captures broader questions about the inductive bias of LMs trained solely on text, and the latent concepts they acquire over the course of training (**Q3**).

Next, by extending the SCM beyond the data generation process to cover the training of the LM (unsupervised) and probe (supervised), we further show that **Q1** and **Q2** can be understood as the mediating and moderating effects of the probe, respectively. We propose a general technique based on *causal mediation analysis* which isolates the causal path through the LM while excluding the probe's influence. Our analysis yields clear, testable conditions for accepting or rejecting our hypotheses based on a probing experiment's outcomes.

Finally, we conduct an empirical study that extends Jin & Rinard (2024), who use probes to quantify the extent to which LMs are capable of learning "meaning" from text, as operationalized by the semantics of a synthetic programming language for grid-world navigation. By leveraging the proposed **latent causal probing** framework, our experiments allow us to draw precise conclusions about the *causal relationship* between the latent dynamics that generated the training data and what is learned by the LM. In particular, we find evidence that (1) the LM has, in fact, learned to represent the latent variables corresponding to the underlying semantics of the language, and (2) the LM representations exhibit an inductive bias that generalizes to novel action sequences. Our study marks the first rigorous empirical evaluation of the hypothesis that *language models are latent concept learners*, revealing intriguing insights into how language models might acquire an understanding of language. Code to reproduce our experiments is available at `https://github.com/charlesjin/emergent-semantics`.

## 2 Structural causal models of text

This section introduces the setting of our framework for probing, which is based on the idea that the text used to train LMs may exhibit latent causal structure; we formalize these concepts using the approach of structural causal models.

### 2.1 Background: structural causal models

Structural causal models are graphical models which represent causal relationships in a data generation process as directed graphs (Pearl et al., 2000). We refer the reader to Pearl (2010) for a comprehensive overview. Suppose that we are interested in the effect the weather has on employees bringing an umbrella to work. In this case, we may hypothesize a SCM like the one in Figure 1a. Each node represents a random variable: the weather, the weather forecast, whether the employee's morning gets off to a late start, and whether the employee brings an umbrella to work. Nodes without parents are *exogenous* variables, whose causes are left unexplained; they are often used to model nature or randomness. Nodes with a

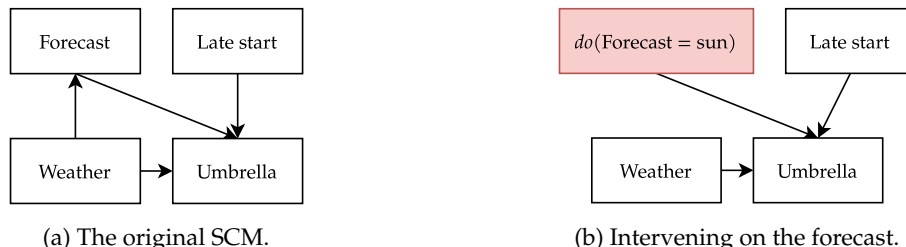

(a) The original SCM.                    (b) Intervening on the forecast.

Figure 1: An SCM for bringing an umbrella to work.

parent indicate the possibility of a causal relationship, e.g., the edge from weather to forecast indicates that the weather might influence the forecast. A standard assumption of causal analysis is that the underlying causal graph is Markovian (or acyclic).

**Mediators and moderators.** The SCM hypothesizes 3 possible causes for how the weather affects employees bringing an umbrella to work: the weather, the weather forecast, and having a late start. The forecast is a *mediator* because total causal effect of the weather on umbrella in partially transferred by the *path-specific effect* over the weather-forecast-umbrella pathway (Avin et al., 2005; Imai et al., 2010). A natural question is how much the forecast is responsible for the increase in likelihood that an employee brings an umbrella to work when, for instance, the weather changes from sunny to rainy. This can be analyzed via *necessary indirect effects*, which quantify how much the presence of the causal path through the mediator contributes to the total measured effect (Weinberger, 2019):

$$\text{NIE}_{\text{rain,sun}}(\text{Forecast}) = \mathbb{E}[\text{Umbrella} \mid \text{Weather} = \text{rain}]$$
$$- \mathbb{E}[\text{Umbrella} \mid \text{Weather} = \text{rain}, do(\text{Forecast} = \text{sun})],$$

where $do(\text{Forecast} = \text{sun})$ is a *causal intervention* that can be conceptualized as forcing the weather station to forecast sun regardless of the weather, thereby severing the weather-forecast-umbrella pathway. Figure 1b depicts the SCM post-intervention.

The late start variable is a *moderator* of the weather-umbrella causal effect: variables that do not directly mediate a causal effect, but affect the strength (and possibly direction) of another causal path (Baron & Kenny, 1986). For instance, the forecast's effect (i.e., $\text{NIE}_{\text{rain,sun}}(\text{Forecast})$) might be lower if the employee has a late start and rushes out the door without checking the forecast.

### 2.2 Case study: causal structure in programming languages

Jin & Rinard (2024) propose an experiment to study whether LMs are able to ground a sequence of actions into a sequence of states, having only seen instances of the initial and final state during training. Specifically, they train a 350M parameter Transformer (Vaswani et al., 2017) on a corpus of *specification-program* examples using a standard causal language modeling objective. The programs are strings in a grid-world navigation language with 5 actions (move, turn_right, turn_left, put_marker, pick_marker), sampled uniformly between lengths 6 and 10, inclusive. The specifications consist of the initial and final state, which are 8x8 grids. Executing the program navigates a single robot in the initial state to the final state. We refer to Jin & Rinard (2024) for more details about the language.

Figure 2 displays an SCM of the data generation process (along with an example assignment of values to each variable). The exogenous variables are the initial state and the program actions. Each action produces a latent state (green), save for the last action, which is observed as the final state. A training sample consists of the sequence: $s_0, s_n, p_1, \ldots, p_n$, where each grid world is converted to text by scanning in row order, with one token per entry.

Consider now modeling a distribution of text drawn from this SCM. In particular, for each sample $x$ there is an assignment $e$ to the exogenous variables in the SCM $M$ such that $M(e) = x$. One strategy would be to learn a model of the SCM depicted in Figure 2, and

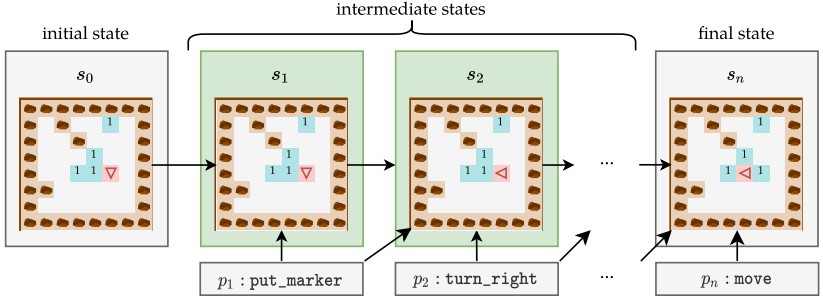

Figure 2: An SCM of the data generation process for the grid world corpus. The exogenous variables are the initial state and the actions; latent variables are green and observed variables are gray. The training corpus consists of programs of length between 6 and 10.

integrate the latent variables during inference. For instance, knowing that the robot is one space away from $s_n$ in $s_{n-1}$ could help a learner predict $p_n =$ move.

More generally, given observations generated according to some unknown causal mechanism, a learner could propose various SCMs of the underlying causal mechanism consistent with the observations, then use these SCMs to inform future predictions, an approach known as causal learning (Schölkopf et al., 2021). A major challenge in the foregoing approach is the problem of *latent variable induction*, or inferring the latent variables over which candidate SCMs are to be defined. In this work, we focus on the causal structure of programming languages, where the underlying causal dynamics are governed by a precise *formal semantics*, and the latent variables are given by program states. Having formally defined semantics and latent variables enables us interpret the results of our probing experiments in an unambiguous way; we refer the reader to Sloman (2005); Feder et al. (2022) for surveys of causal structure in natural language.

## 3 Latent causal probing

We present **latent causal probing**, a formal framework for empirically testing the hypothesis

*Language models are latent concept learners*.

At a high level, given an SCM that models the training data as the observed variables, we probe the LM for representations of the latent variables of the SCM. Our main insight, as illustrated in Figure 2, is that knowing the latent value of $s_{n-1}$ could help predict the observed value of $p_n$; hence, an LM trained to predict $p_n$ might eventually induce the existence of the latent variable $s_{n-1}$.

### 3.1 Probing for latent concepts

We begin by defining the auxiliary task and dataset for probing. Fix some structural causal model $M$, and let $v_M$ be the latent variable of interest. Given some text $x$, we use $v_M(x)$ to denote the value of the latent variable in the SCM of text $x$. For instance, the value of $v_M = s_1$ in the sample $x$ from Figure 2 is the grid depicted in the $s_1$ node. We assume that the value of each latent variable is uniquely determined by $x$ and $M$.

Given a language model $LM$ with parameters $\theta$, we denote an arbitrary representation function as $LM(x; \theta)$. The auxiliary dataset consists of input features $\{LM(x; \theta) \mid x \in X\}$ and labels $\{v_M(x) \mid x \in D\}$, where $D = \{x_i\}_{i=1}^N$ is a corpus of text. We then split $D$ into two auxiliary datasets: one for **calibration** and one for **measurement**. The probe is trained to predict $v_M(x)$ given $LM(x; \theta)$ on the calibration data, and the accuracy is taken over the measurement data. We next discuss the design and interpretation of these two datasets.

| calibration | measurement | |
| --- | --- | --- |
| | bound | free |
| bound | deductive knowledge | inductive bias (inference) |
| free | deductive bias (consistency) | inductive knowledge |

Table 1: Interpreting probing with different calibration and measurement datasets.

**Bound vs. free latent variable outcomes.** In general, there may exist several causal dynamics that explain the data equally well. For instance, the following dynamics could also generate the data in Figure 2:

`put_marker` Jump to a random location.

`turn_right` Return to the last position, put a marker, then turn right.

These dynamics assign a different value to $s_1$, but explain the observed variables equally well. Assuming the training corpus consists entirely of this single example, it would be impossible to distinguish between $M$ and $M'$ on the basis of data alone. In other words:

1. $M$ and $M'$ share the same set of set of latent, observed, and exogenous variables;

2. $M$ and $M'$ agree on the observed data; and

3. there exists an assignment $e$ to the exogenous variables such that $v_M(x) \neq v_{M'}(x')$ for $x = M(e)$ and $x' = M'(e)$.

In this case, we say that the latent variable $v$ is **free** over the assignment $e$. More generally, given a hypothesis class $\mathcal{M}$ of SCMs over the same set of variables, denote the LM training data as $D_{\text{train}}$ and define $\mathcal{M}|_{\text{train}}$ to be the subset of SCMs that generate $D_{\text{train}}$. The free latent variable outcomes consist of pairs of latent variables and assignments $(v, e)$ such that there exist $M, M' \in \mathcal{M}|_{\text{train}}$ where $v_M(M(e)) \neq v_{M'}(M'(e))$. Any latent variable outcome $(v, e)$ which is not free is **bound**, i.e., its value is fully determined by the training data.

**Probing with free vs. bound splits.** Table 1 details four possible probing setups when separating the auxiliary dataset $D$ into free and bound splits. In particular, when calibration and measurement occur on the same split, the probe quantifies the **knowledge**, or information content, that can be extracted from the LM representations; conversely, probing with different splits measures the transferability of the representations across different splits, which is a **bias**. Additionally, because the bound variables outcomes, can, by definition, be deduced from the given data (and hypothesis class $\mathcal{M}$), measuring on the bound split relates to the **deductive** ability of the LM; conversely, measuring on the free split is inherently an **inductive** process. We highlight that the inductive bias can be understood as quantifying the capacity of the LM representations to *infer values in unseen data by applying theories derived from known data*, a form of inductive inference; while the deductive bias measures the extent to which the LM representations *produce theories of unseen data that are consistent with the observed data*, a key tenet of deductive logic.

### 3.2 Causal mediation analysis of probing

We next turn to controlling for the probe (**Q1**). Intuitively, the challenge is any measurement using a supervised probe conflates the LM's representation of the auxiliary labels with the probe's ability to learn the auxiliary task (Hewitt & Liang, 2019). While there exist a number of proposals for controlling for the contribution of the probe, such techniques typically do not provide any formal guarantees, rendering their correct application and interpretation a challenge (Belinkov, 2022).

We propose a method of disentangling the two effects using the formal framework of causal mediation analysis, and specifically, path-specific effects, which analyze how causal effects decompose over multiple causal paths (Avin et al., 2005; Imai et al., 2010). To begin, we extend the SCM of the data generation process to include (1) the LM training, (2) the probe

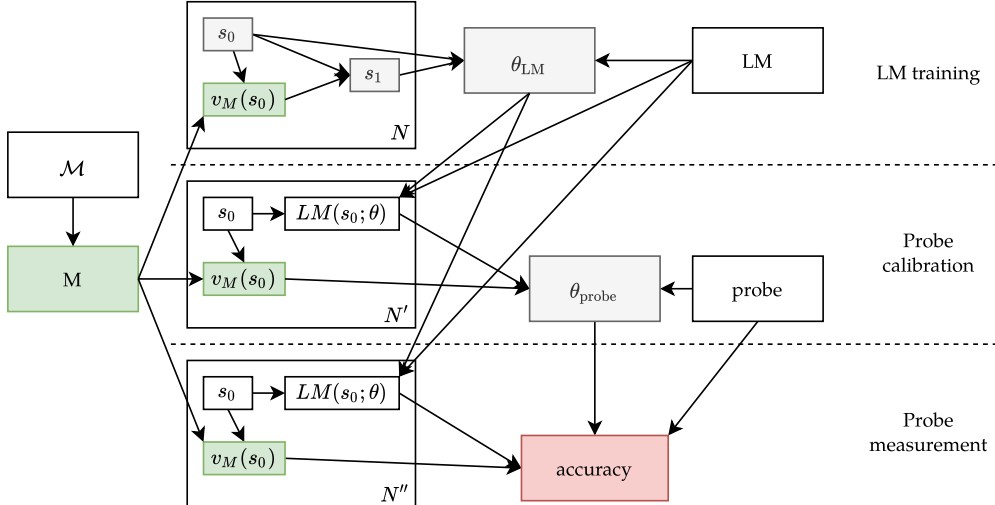

Figure 3: An SCM depicting the LM training, probe calibration, and probe measurement. We use plate notation for repeated iid samples, e.g., we draw $N$ samples for LM training.

calibration, and (3) the probe measurement. Figure 3 illustrates an example where the hypothesis class $\mathcal{M}$ consists of 3-variable SCMs with a single exogenous, observed variable $s_0$, a latent variable $v_M$, and another observed variable $s_1$.

Observe that there are three causal paths from the true SCM of the data generation process to the auxiliary task accuracy, each of which is mediated by a different set of the latent variables $v_M$: (1) during LM training, the LM is trained on a dataset whose text is causally affected by $v_M$; (2) during probe calibration, the probe is calibrated using $v_M$ directly; and (3) during probe measurement, the probe is evaluated for accuracy on $v_M$ directly. However, we only care to measure the effect over the first of these causal paths, i.e.:

*To what extent can the auxiliary task performance be attributed to what LM learns from the latent variables in its training data?*

This question can be posed formally as the necessary indirect effect of the paths mediated by the LM's learned representation for some baseline causal dynamics $M'$:

$$\text{NIE}_{M,M'}(\theta_{\text{LM}}) =$$
$$\mathbb{E}\big[\text{accuracy} \mid \text{LM is trained on } M, \text{ probe is calibrated and measured on } M\big]$$
$$- \mathbb{E}\big[\text{accuracy} \mid do(\text{LM is trained on } M'), \text{ probe is calibrated and measured on } M\big],$$

Although path-specific effects offer a crisp conceptual framework for isolating the contribution of the LM in probing experiments, actually computing NIE is not straightforward. First, picking a proper baseline $M'$ is critical: if we pick an inappropriate $M'$, then the NIE will measure the difference between the data generated by $M$ and $M'$ in addition to the latent variables hypothetically mediated by the LM representations. For instance, we could have $M'$ produce gibberish text. Then the LM wouldn't learn anything, biasing the NIE to be positive. Second, measuring the effect requires training a new LM with the baseline $M'$, which would be prohibitively expensive for large pretrained LMs.

Let $acc_{aux}(M_0, M_1)$ denote the (expected) auxiliary task accuracy after the LM is trained using the SCM $M_0$ and the probe is calibrated and measured on $M_1$. The following result addresses these challenges (proof in Appendix C).

**Definition 3.1** (Valid baseline). *$M'$ is a **valid baseline** for M if*

$$acc_{aux}(M', M') \geq acc_{aux}(M, M) \tag{1}$$
$$acc_{aux}(M, M') \geq acc_{aux}(M', M). \tag{2}$$

**Proposition 3.2.** *Let $M'$ be a **valid baseline** for M. Then*

$$acc_{aux}(M, M) - acc_{aux}(M, M') > 0$$

*implies both $NIE_{M,M'}(\theta_{LM}) > 0$ and $NIE_{M',M}(\theta'_{LM}) > 0$.*

Intuitively, $M'$ is a valid baseline when measuring $M'$ is easier than measuring $M$ under both normal or intervened circumstances. The conclusion then states that, so long as $acc_{aux}(M, M) - acc_{aux}(M, M') > 0$ (which can be evaluated by running probe calibration and measurement twice rather than training the LM twice), there is no bias in which SCM is used to train the LM and which is the baseline: the LM representations always mediate a positive amount of the measured effect. Intuitively, given (1) an LM trained on either $M$ or $M'$ and (2) some neutral text $x$ that is equally likely to have been generated by either $M$ or $M'$, we could distinguish which data the LM was trained on purely on the basis of which latent concepts it assigns to the neutral text $x$. A positive NIE now also has a rigorous interpretation as the LM having *induced latent concepts*, as some positive amount of causal effect is transferred through the representations of the LM. For instance, a positive mediated measurement for inductive bias implies that

*The presence of latent causal variables in the pretraining data causes the LM to learn representations that generalize to unknown data.*

### 3.3 Discussion

We summarize the **latent causal probing** framework as follows:

1. Fix the exogenous, latent, and observed variables and design a hypothesis class $\mathcal{M}$.
2. Pick a target SCM $M \in \mathcal{M}$ and a set of latent variables $v \in M$ to test.
3. Construct the auxiliary dataset and create the bound vs. free splits (if possible).
4. Identify a valid baseline $M'$ and perform the mediation analysis.

A significant measurement $acc_{aux}(M, M) - acc_{aux}(M, M') > 0$ is interpreted evidence that the LM encodes the latent concepts in its representations. We conclude with some remarks.

**Choice of hypothesis class.** One requirement of our framework is to pick an explicit hypothesis class $\mathcal{M}$, which yields a separation of the latent variables into bound vs. free splits. Conceptually, the choice of hypothesis class forces the experimenter to make explicit their prior about what a language model can "deduce" from data. In our case, we will assume that language models can perfectly memorize the behavior of previously seen programs, but cannot extrapolate this information to any previously unseen programs. Our construction of the hypothesis class reflects this assumption, and the deductive knowledge measures the ability of the language model to directly recall this information from the training data.

**Interventions, and probing for non-causal latent variables.** Our mediation technique requires knowing "what would the text have been if the underlying dynamics were different?", which could be difficult (especially in non-synthetic domains). Similarly, for non-causal latent variables, such as part-of-speech, producing a hypothesis class $\mathcal{M}$ with more than one SCM may seem unnatural: what would the data look like in a counterfactual world in which "dog" is actually an adverb? Our analysis suggests that a baseline $M'$ which induces a different distribution of text is a necessary precondition, since otherwise $NIE_{M,M'}(\theta_{LM})$ and $NIE_{M',M}(\theta'_{LM})$ cannot both be positive (as $M$ and $M'$ are indistinguishable when used to train the LM and hence $\theta_{LM} = \theta'_{LM}$). Intuitively, we interpret this result as saying *any measurement is inherently biased when the auxiliary task has only one "right" answer*.

**Probe architecture and hyperparameters.** Our framework also explicates the role of the probe's architecture and other hyperparameters in the training process, such as the optimizer, learning rate, dataset size, etc., as potential *moderators*, but not mediators (**Q2**). In other words, so long as there exists hyperparameters such that the NIE is positive, the analysis concludes that there exists a causal effect mediated by the model's parameters (although crucially, these choices must not introduce new confounders). Practically speaking, our

framework also offers a novel way to interpret (and justify) complex probes (Voita & Titov, 2020; Pimentel & Cotterell, 2021).

# 4 Experiments

We conduct an empirical study of whether an LM, trained from scratch on a corpus of program data, induces the latent concepts in the underlying data generation process. We obtained the original LM checkpoints from Jin & Rinard (2024). As the LM achieves 92.4% accuracy on generating semantically correct programs for unseen specifications by the end of training, we consider the LM to be *well trained* in the sense that it has successfully fit not only the explicit objective of minimizing the next-token prediction loss, but also the implicit task encoded by the SCM, i.e., generating programs that correctly implement specifications. We thus evaluate the hypothesis that a well trained language model is a latent concept learner. Appendix A.2 contains further experiments and analyses, including two additional valid baselines and a study of the moderating effect of the probe architecture and training.

## 4.1 Methods

We describe the key steps according to the framework in Section 3.3; Appendix A.1 contains full experimental details (e.g., LM and probe architecture and training, LM representations).

**Hypothesis class.** The exogenous variables are the initial state and program. The latent variables are the intermediate states, and the observed variables are the initial and final state and the program. For the hypothesis class $\mathcal{M}$, we will assume that each program is a purely symbolic string that represents a sequence of instructions, but that the instructions are not necessarily compositional. For instance, the string "turn_right" could map to the function [[move move]] while the string "turn_right turn_left" could map to the function [[move]]. Then to execute the program "turn_right turn_left" on an initial state $s_0$, the robot would first execute "turn_right" and visit $s_1 = \text{move}(\text{move}(s_0))$, and then execute "turn_right turn_left" which would result in the final state being $s_2 = \text{move}(s_0)$. Note that the *function* represented by each string can be inferred from input-output examples, but that the intermediate states in the *execution* cannot.

**Target SCM and latent variables.** The target SCM $M \in \mathcal{M}$ is the true data generation process in Figure 2. The target latent variables consist of the robot's position, facing direction, and whether the robot is facing a rock for each intermediate state.

**Auxiliary dataset splits.** For the auxiliary dataset, we use the same data generation process, except that programs range in length between 1 and 15, and we replace the final state in the specification with the initial state. We assume the LM observes all combinations of the exogenous variables. The bound latent variables are $s_6$ to $s_{10}$ (they correspond to the final state of programs observed during training). The free latent variables are $s_1$ to $s_5$ and $s_{11}$ to $s_{15}$. Note that we overapproximate the bound latent variables (the input-output behavior for some programs of length 6 to 10 might not be completely known given the training data), but the free latent variables are guaranteed to be free (i.e., the training data does not contain any input-output behavior for programs of length less than 6 or greater than 10).

**Valid baseline.** We construct a valid baseline by using the same causal structure as the target SCM, but permuting the causal dynamics of the turn_right, turn_left, and move actions (e.g., the robot turns left when executing a turn_right action). Note that the valid baseline still has compositional semantics. As $M$ and $M'$ are clearly symmetric from a language modeling perspective, Definition 3.1 (and hence Proposition 3.2) holds.

## 4.2 Results

Figure 4 plots the main results. For all four measurements (deductive knowledge, inductive bias, deductive bias, and inductive knowledge) and across all three probes (linear, 1-layer MLP, 2-layer MLP), the mediated measurements are significantly positive (each green region is entirely above the corresponding red region) by the end of training. We thus conclude

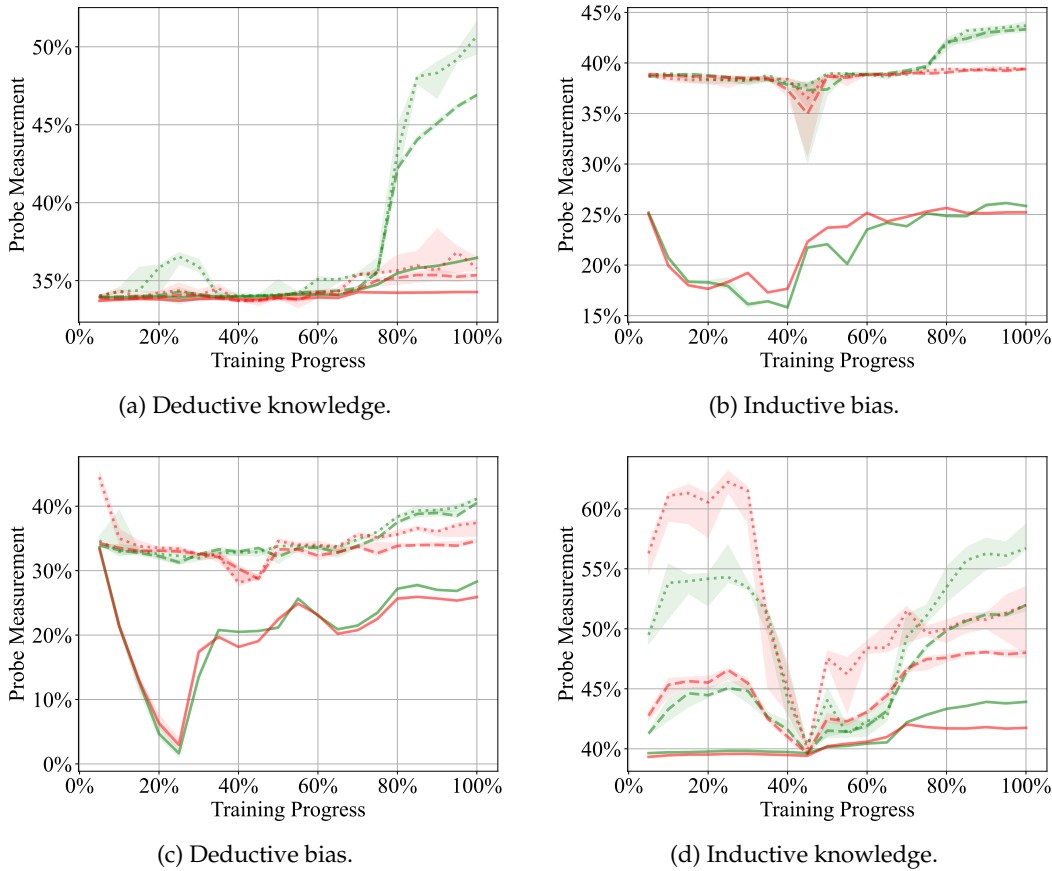

Figure 4: Experimental results. Solid, dashed, and dotted lines plot the median accuracy of a linear, 1-layer MLP, and 2-layer MLP probes, respectively. Each shaded area represents the full range over 5 random seeds. Green is the target SCM and red is the baseline.

that a positive fraction of the *observed measurements of latent concepts can be attributed to what is learned by the LM's representations*. We make several additional observations:

**Spurious features dominate early training.** Early in training, both raw and mediated measurements show significant variance and can even decrease with additional training. For example, the mediated measurement for inductive knowledge starts high for all three probes but drops to a minimum around halfway through training. Furthermore, for both the deductive bias and inductive knowledge, the highest raw measurement over the entirely of training is achieved in the first quarter of training by the baseline, rather than the target, SCM. Finally, when using the linear probe to measure the deductive bias, the highest raw measurement for the target SCM does not even occur at the end of training, but at the beginning. We attribute such behavior to *spurious features*, which are not the result of training but rather exist in random initialization of the LM parameters. These observations suggest that (1) spurious features early in training can lead to uninformative results for probing experiments, and (2) LMs can unlearn spurious features *even when correlated with the target SCM*, before relearning them again later in training.

**Deeper probes are (generally) more accurate.** For almost every raw measurement across the entirety of training, deeper probes exhibit better auxiliary accuracy. Furthermore, at the end of training, the linear probe exhibits the lowest mediated measurements while the 2-layer MLP exhibits the highest mediated measurement in two of the four tasks. Only in the case of the deductive bias does the 1-layer MLP achieve a substantially higher measurement than the 2-layer MLP. The improved mediated measurements suggests that probes with greater capacity can yield clearer signals and, more generally, highlights the importance of probing frameworks that can robustly account for the design of different probes.

**Raw measurements can be biased.** At the end of training, the raw measurements for inductive knowledge are approximately 52% and 58%, compared to 48% and 51% for deductive knowledge, using a 1-layer and 2-layer MLP, respectively. However, for the mediate measurements, the inductive knowledge shows roughly 4% and 7%, while deductive knowledge shows 12% and 15%, respectively. This reversal demonstrates how raw measurements can be confounded by how easy or hard the auxiliary task is for the probe to fit.

## 5 Related work

**Causal interpretability of LMs.** Several prior lines of work apply causal techniques to the interpretability of LMs. These works typically intervene on either the model's representations (Elazar et al., 2021; Geiger et al., 2021; Meng et al., 2022; Abraham et al., 2022; Li et al., 2022) or the model's inputs (Kaushik et al., 2020; Vig et al., 2020; Gangal & Hovy, 2020; Amini et al., 2023), and analyze the causal effect on the LM's outputs. In contrast, we present a formal framework that, conceptually, intervenes on the model's *training data*, and measures the causal effect on the LM's internal representations. Elazar et al. (2022) also study the causal relationship between the LM and its training data, but they focus on the causal effect of dataset statistics on the factuality of LM's outputs.

**World models in LMs.** The hypothesis that LMs can induce latent concepts is related to evidence of world models in LMs, or the extent to which LMs are capable of grounding their inputs to (some representation of) reality. For instance, Li et al. (2021) find that LMs perform entity state tracking over the course of simple stories. Li et al. (2022) show that an LM trained on Othello transcripts develops a representation of the underlying board state. Our work is based on Jin & Rinard (2024), who show that an LM can develop representations of the intermediate world states underlying a sequence of instructions, given only instances of the initial and final states. However, none of these works provide theoretical guarantees for controlling the classifier's ability to fit the auxiliary objective. This leaves open the possibility that the LM might simply represent the text as is, with either the text directly encoding the world model or the probe learning to infer the latent world model from the text. Our formulation of world models according to the underlying data generation process is also highly related to the position developed by Andreas (2022), who argues that LMs could act as "agent models" that model properties of agents that are likely to have generated the language in their training data.

**Frameworks for probing.** A number of works have proposed frameworks toward a more rigorous understanding of probing. One line takes an information-theoretic view on the information represented by the LM (Zhu & Rudzicz, 2020; Pimentel et al., 2020; Voita & Titov, 2020; Pimentel & Cotterell, 2021). Our work leverages causal analysis to attribute the presence of knowledge and biases in the LM's learned representations. Immer et al. (2022) propose an interpretation of probing as quantifying the inductive bias of pretrained representations for downstream tasks, but their framework differs significantly from ours in that the model is understood as a representation-probe pair. In contrast, our approach explicitly aims to separate the LM training and probe calibration procedures using causal mediation analysis, and interprets probes as a means of measuring causal effects. Our analyses also reveals settings in which prior techniques, such as control tasks (Hewitt & Liang, 2019) and the use of untrained or otherwise random baselines (Zhang & Bowman, 2018; Belinkov, 2022), can yield misleading estimates of the intended auxiliary measurement.

## 6 Conclusion

This paper presents **latent causal probing**, a probing framework that studies whether LMs induce latent variables as a byproduct of the language modeling objective. Our framework offers robust tools for interpreting experiment results through the lens of causal analysis, and in particular, rigorously controls for the probe's contribution in learning the auxiliary task. Experimentally, we extend a previous study of whether Transformers can infer the intermediate states that underlie a sequence of actions. Our results provide strong empirical evidence that LMs can induce latent concepts from textual pretraining.

## Acknowledgements

We sincerely thank Jacob Andreas, Armando Solar-Lezama, and the anonymous COLM reviewers for their insightful feedback on earlier drafts of this manuscript. We also gratefully acknowledge support from DARPA Grants HR001120C0015, HR001120C0191, and N6600120C4025. The views expressed are those of the authors and do not reflect the official policy or position of the Department of Defense or the United States Government.

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

# A  Additional experimental details and results

## A.1  Language model and probe details

Following Jin & Rinard (2024), the language model is a 350M parameter CodeGen model (Nijkamp et al., 2023) taken from the HuggingFace Transformers library (Wolf et al., 2020). The model was trained for 2.5 billion tokens, which was roughly 6 passes or 80000 training batches over the training corpus. We refer to Jin & Rinard (2024) for further details.

We next describe the design and training of the probing classifiers; these notes apply to all the probing experiments, unless otherwise noted. The linear probe is a single linear layer. The MLP probes have ReLU, batch_norm, then dropout(p=.2) after each linear layer. The hidden dimensions of the 1-layer and 2-layer MLP probes were (256,) and (256, 1024), respectively. The auxiliary datasets consisted of 500000 randomly selected samples. To extract representations from the LM, we use the same strategy as Jin & Rinard (2024), averaging the LM hidden states over the layer dimension after processing each program token. Probes were trained using AdamW (Loshchilov & Hutter, 2019) with weight decay of 1e-4. The learning rate starts at 0.01, then decays by .1 at 75% and 90% through training. All probes are trained for 2000000 steps using a batch size of 256.

For the mediated results reported in Figure 4, we generated the auxiliary dataset using an SCM that maps `turn_right` to `turn_left`, `turn_left` to `move`, and `move` to `turn_right`.

## A.2  Ablation studies

This section present some ablation studies on the set up of the probing experiments.

### A.2.1  Valid baseline selection

To test the sensitivity of the mediated results (and hence, overall conclusions) on the choice of valid baseline, we generate two additional auxiliary datasets with the following SCMs:

1. swap `move` and `turn_left`
2. swap `turn_right` and `turn_left`

The results are plotted in Figure 5 and Figure 6, respectively. We find that the mediated measurements in the first case are nearly identical to those in Figure 4, despite only swapping two actions (instead of permuting three). However, in the second case, the mediated measurements are essentially noise, with almost complete overlap between the target and baseline auxiliary measurements. We attribute this to the fact that the resulting labels are extremely similar, as, in most cases, the robot is simply reflected along the starting axis.

We emphasize that, strictly speaking, no amount of negative empirical results can definitively prove that LMs are *never* capable of latent concept learning. Conversely, a single positive result with a valid baseline does constitute evidence that LMs can *sometimes* induce latent concepts from text.

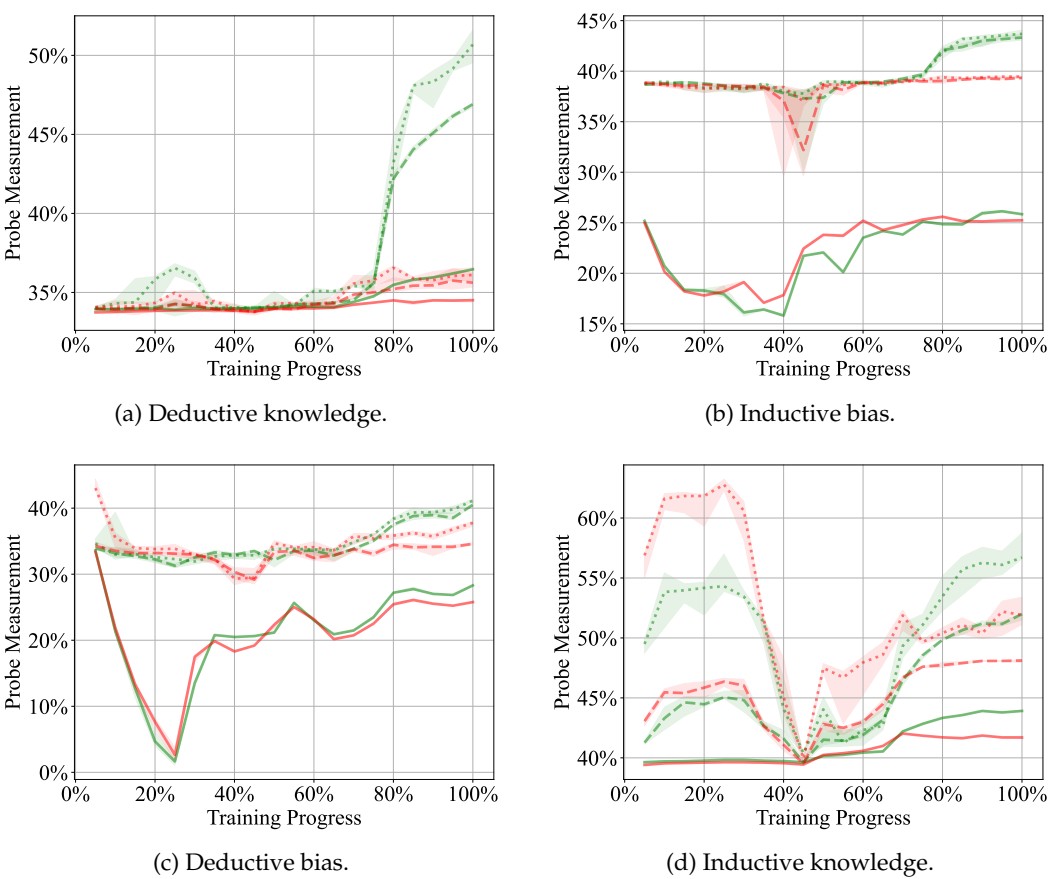

(a) Deductive knowledge.

(b) Inductive bias.

(c) Deductive bias.

(d) Inductive knowledge.

Figure 5: Mediating with the valid baseline that swaps move and turn_left.

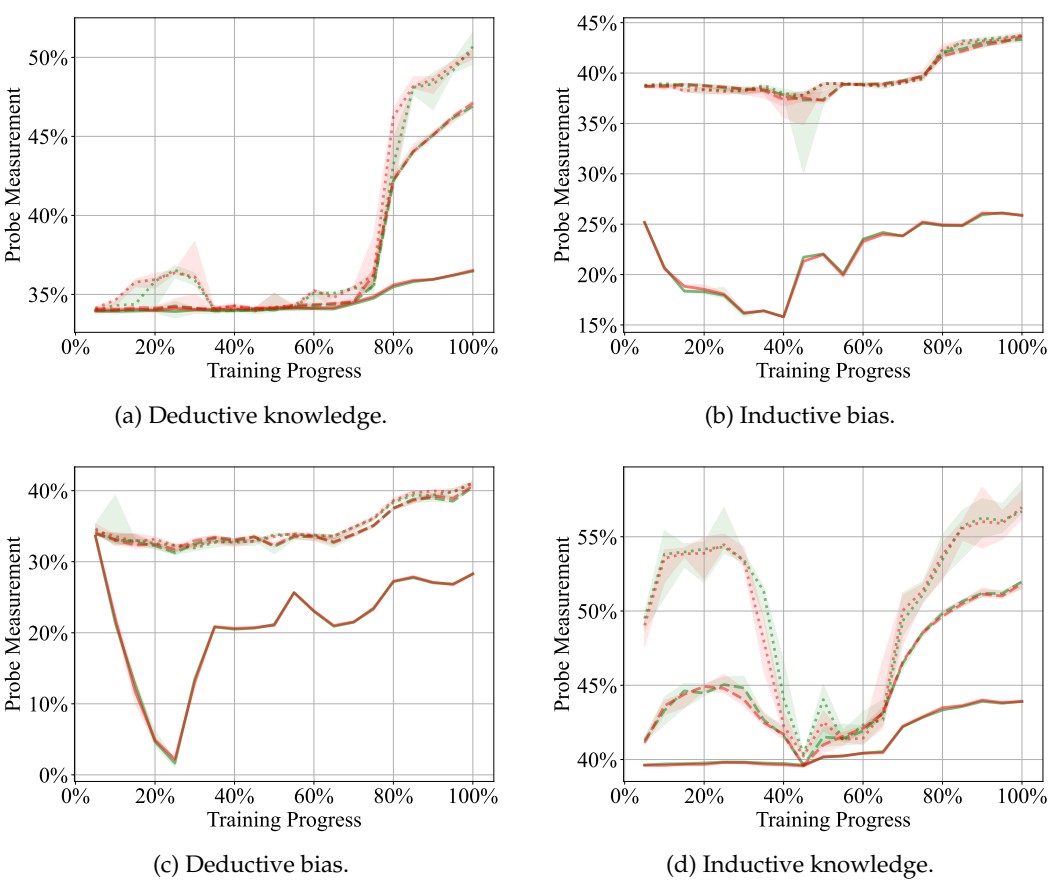

(a) Deductive knowledge.

(b) Inductive bias.

(c) Deductive bias.

(d) Inductive knowledge.

Figure 6: Mediating with the valid baseline that swaps `turn_right` and `turn_left`.

### A.2.2 Probe architecture and hyperparameters

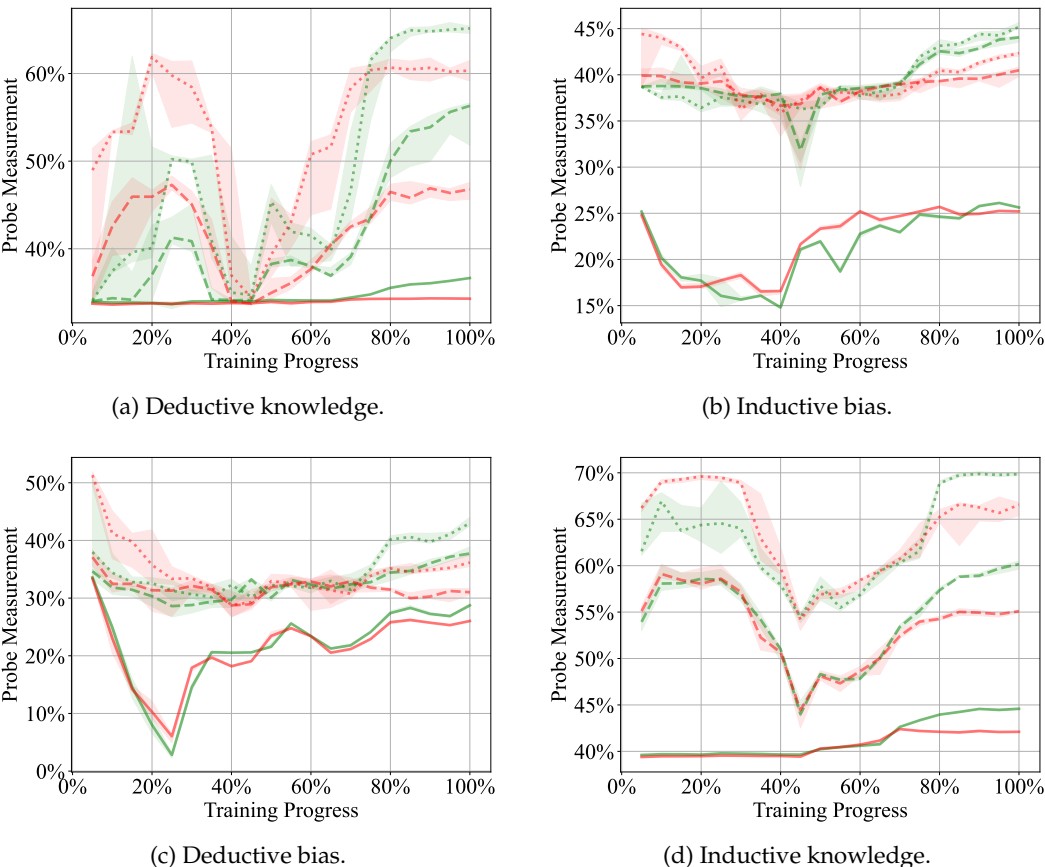

Figure 7: Mediating with the valid baseline from the main text. Both the original and the mediated measurements are retaken using the probe architecture and hyperparameters from Jin & Rinard (2024).

We next ablate the probe architecture and hyperparameters by adopting the settings used in Jin & Rinard (2024). The differences are: no dropout, a batch size of 1024, training the probe for 10000000 steps, and using 100000 samples in the auxiliary dataset. We use the same valid baseline as in Figure 4 of the main text.

The results are plotted in Figure 7. We observe that the general trends are preserved, and all four target measurements are above the baseline measurements by the end of training (i.e., the mediated measurements are positive). However, we note that both deductive and inductive knowledge measure slightly lower, which is an example of the moderating effect of the probe architecture and training hyperparameters. We attribute the effect to the increased batch size and lack of dropout, which could encourage the probe to converge more quickly to a global optimum, given that the risk of overfitting is low (due to the large size and high quality of the training dataset). This is also consistent with (1) the general intuition that simpler (or less optimal) probes are a proxy for "ease of extraction," which is often interpreted as evidence that the representations are "more aligned" with the target features (Hewitt & Liang, 2019), and (2) the theoretical findings in Pimentel et al. (2020), who conclude that probes of infinite capacity are most informative for measuring (syntactic) knowledge.

## B  Comparison with Jin & Rinard (2024)

In this section, we highlight several key departures from the experimental design in Jin & Rinard (2024).

First, they do not split their auxiliary dataset into bound and free latent variables, and hence their results do not yield fine-grained interpretations about probing with different calibration and measurement datasets.

Second, our analysis reveals the presence of possible confounders in the design of their interventional baseline, leading to *uncontrolled effects*. In particular, the auxiliary dataset is constructed using programs generated by the LM itself, rather than randomly sampled as we do. Intuitively, this means that the LM "sees" both $s_0$ and $s_n$, *which reveals information about the original casual dynamics*. Formally, the representations of the LM used for probing mediates all 3 causal pathways, rather than the simple causal pathway from the LM training data (as in Figure 3), and hence their interventional baseline is not a proper measurement of the causal effect mediated by the LM representations. Our solution is to use randomly sampled programs and replace the occurrence of $s_n$ with $s_0$ in the construction of the auxiliary dataset, which breaks this causal dependence.

Finally, Jin & Rinard (2024) do not verify that their interventional baselines satisfy the conditions in Equations (1) and (2). In particular, one of their baselines map the `put_marker` and `pick_marker` actions to `turn_right` and `turn_left`, respectively, in addition to permuting the `turn_right`, `turn_left`, and `move` actions. Because the extracted features all relate to the position and direction of the robot, the new dynamics could present a more difficult task (for both the LM and the probe) due to replacing what were effectively no-ops (`put_marker` and `pick_marker`) with new operations that affect the position or direction (`turn_right`, `turn_left`, and `move`). Hence, the observed drop in accuracy post-intervention could be attributable to increased task difficulty, rather than the learned representations of the LM.

## C  Proofs

*Proof of Proposition 3.2.*  The proof follows directly from substituting the appropriate assumptions into the definitions of NIE. Recall that

$$\text{NIE}_{M,M'}(\theta_{\text{LM}}) := acc_{aux}(M, M) - acc_{aux}(M', M) \tag{3}$$

$$\text{NIE}_{M',M}(\theta'_{\text{LM}}) := acc_{aux}(M', M') - acc_{aux}(M, M'), \tag{4}$$

and, by Definition 3.1, $M'$ is a valid baseline for $M$ if

$$acc_{aux}(M', M') \geq acc_{aux}(M, M) \tag{5}$$

$$acc_{aux}(M, M') \geq acc_{aux}(M', M). \tag{6}$$

Applying Equation (5) to the definitions of NIE,

$$\text{NIE}_{M,M'}(\theta_{\text{LM}}) \leq acc_{aux}(M', M') - acc_{aux}(M', M) \tag{7}$$

$$= \text{NIE}_{M',M}(\theta'_{\text{LM}}). \tag{8}$$

Applying Equation (6) to the definition of NIE,

$$\text{NIE}_{M,M'}(\theta_{\text{LM}}) = acc_{aux}(M, M) - acc_{aux}(M', M) \tag{9}$$

$$\geq acc_{aux}(M, M) - acc_{aux}(M, M'). \tag{10}$$

Hence,

$$acc_{aux}(M, M) - acc_{aux}(M, M') \leq \text{NIE}_{M,M'}(\theta_{\text{LM}}) \leq \text{NIE}_{M',M}(\theta'_{\text{LM}}) \tag{11}$$

$$\square$$

