# OpenReview forum: "Latent Causal Probing: A Formal Perspective on Probing with Causal Models of Data"
_colmweb.org/COLM/2024/Conference — COLM_

### Official Review · Reviewer_B6Ky · 2024-04-28

**Rating:** 6
**Confidence:** 4
**Ethics Flag:** 1

**Summary:**

The paper proposes a formal probing framework to test whether LMs have learned the right laten variables to behave as a “structural causal model” (SCM). An SCM forms a directed graph of  token representations as a causal pathway from an initial state to a final state. The proposed methods tries to separate what is learned during pretraining (LM) from the training (calibration) for probing (SCM) by differentiating free (outside the calibration) and bound (seen during pretraining and calibration) variables  Their method is tested using a Code-generation model that is trained on a corpus of specification-program pairs. Their empirical test shows that the LM representation positively contribute to the causal predictive accuracy.

**Reasons To Accept:**

- a probing frame work that goes beyond testing solely on the elicited output but tries to trace the causal relations between implicit latent variables that lead to such output
- provides evidence that the LM learns representations that support such causal structures

**Reasons To Reject:**

- the code generation “language” is very constrained (5 actions) and it is not clear how this can be used for LLMs that learn far more complex and diverse representations that may interfere  in many different ways.
- the section on the results is only 9 lines and way to short. Figure-4 is too small to read and is hardly addressed in the results section. It is not clear what the figure a - h stand for and what the different graph patterns show.
- critical discussion lacks on how this method can be extended to more divers and complex LLMs.

---

> ### Author Rebuttal · Authors · 2024-05-31
>
> Thank you for taking the time to provide us this feedback! We respond to the main points here. If any concerns require further addressing, we would appreciate the opportunity in the discussion period
>
> > the code generation “language” is very constrained (5 actions) and it is not clear how this can be used for LLMs that learn far more complex and diverse representations that may interfere in many different ways.
>
> Our methodology can (in principle) be applied to any language, regardless of how complex the LLM is, so long as you have a model for how the latent variable of interest causally affects the distribution of data.
> We chose a very constrained setting so that we could follow the methodology as faithfully as possible: conducting our experiments under “ideal” circumstances allows us to demonstrate the benefits of the new framework as cleanly as possible and minimize concerns around whether the results could be due to a misapplication of the methodology.
>
> Practically speaking, the underlying causal structure could be less clear in more complex language data. While we agree this makes it difficult to apply the technique to such data, we do want to provide another perspective, which is that our assumptions put the risks of probing on complex data in a mathematically precise language. Specifically, the lack of a clear causal relationship means that we cannot hope for a valid baseline in the exact sense of Proposition 1, so extra care should be taken when interpreting results against a baseline measurement.
>
> > the section on the results ... It is not clear what the figure a - h stand for and what the different graph patterns show.
>
> We will use the extra content page to include an extended discussion of the results in the revision. Briefly, each of the 4 columns shows a different combination of the calibration / measurement split (from left to right: bound / bound, bound / free, free / bound, and free / free, respectively). The top row displays the unmediated results; the bottom row displays the mediated results. The x-axis in all graphs is over LM training.
>
> The most important pattern is that the mediated measurement (bottom) row trends up and positive over the course of LM training, i.e., the LM learns the latent causal variables above a valid baseline.
>
> > critical discussion lacks on how this method can be extended to more divers and complex LLMs.
>
> See our response to point 1 - we will be sure to include this discussion in the paper!

---

> > ### Comment · Reviewer_B6Ky · 2024-06-04
> >
> > Thank you for replying to my review. If the extra space is used to explain and discuss the results more, I would be more inclined to accept the paper. I also propose to discuss more in depth the fact that on true natural language data and models the method may not give conclusive results due to the enormous complexity and variation in causal reasoning that are required for natural language based tasks.

---

> > > ### Author Response · Authors · 2024-06-04
> > >
> > > Thank you for the follow up!
> > >
> > >  >  If the extra space is used to explain and discuss the results more, I would be more inclined to accept the paper.
> > >
> > > We agree this will help make the contributions more clear, and will be sure to dedicate the extra content page to (1) a more in depth discussion of the empirical results and (2) how the methodology can be extended to broader domains (e.g., natural language), including any limitations.

---

### Official Review · Reviewer_nisC · 2024-05-09

**Rating:** 7
**Confidence:** 4
**Ethics Flag:** 1

**Summary:**

This paper proposes latent causal probing, a new probing framework that leverages latent causal structure to design auxiliary tasks. In particular, the task is to assess whether the language model has learned to represent the latent variables of the structural causal model (SCM). A method is proposed to disentangle the causal effects using the causal mediation analysis framework. Theoretical justifications are provided, and empirical results also support the claims of the paper. Overall, the proposed idea is novel, and the paper is well motivated and clearly written.

**Reasons To Accept:**

1. This paper presents a new probing framework from the perspective of causality. The idea is well motivated. The proposed latent probing framework helps address some limitations in existing work.

2. The authors introduced the background of structural causal model in Section 2. Some motivating examples and use cases are discussed and explained.

3. The authors proposed a very interesting idea, i.e., causal mediation analysis of probing.

**Reasons To Reject:**

1. This work only focuses on causal structure of programming languages. It is unclear whether it is easy to extend the proposed latent probing framework to natural language. Additional discussions should be provided in the paper.

2. Experiments only involve an LM trained from scratch. It would be helpful to include evaluations on pretrained LMs.

3. There are some typos in the paper, such as "we refer the ready to" in Section 2.1.

---

> ### Author Rebuttal · Authors · 2024-05-31
>
> We thank the reviewer for finding the paper novel and clearly written! We address the main concerns here, and would be happy to address any remaining questions in the discussion period.
>
> > It is unclear whether it is easy to extend the proposed latent probing framework to natural language. Additional discussions should be provided in the paper
>
> In general, extending latent probing to any new language or domain simply requires having a model of the underlying causal structure. We provide two references to two surveys on the topic of causal structure in natural language (end of section 2.2, page 4); please let us know if you would like further details in the discussion period.
>
> Practically speaking, outside of synthetic datasets, the underlying causal structure of real / natural language data is likely to be much more complex. While we agree this makes it difficult to apply the technique directly on real world data, we do want to provide another perspective, which is that our assumptions put the risks of probing on real world data in a mathematically precise language. Specifically, the lack of a clear causal relationship means that we cannot hope for a valid baseline in the exact sense of Proposition 1, so extra care should be taken when interpreting results against a baseline measurement (see also the discussion in Section 3.3).
>
> We will include a discussion of this point in the extra content page if the paper is accepted.
>
> > Experiments only involve an LM trained from scratch. It would be helpful to include evaluations on pretrained LMs.
>
> We chose to use an LM trained because we felt this would allow us to demonstrate our framework most cleanly with the separation between “free” and “bound” latent variables. For pre-trained LMs, it is impossible to know what the LM actually saw during training (which means it is impossible to know whether a latent variable or concept should be considered free or bound). Training our own LM gives us complete control over the training data, so that we can properly test whether LMs are capable of performing causal induction (i.e., inferring the value of free latent variables vs. just deducing the value of bound latent variables)

---

> > ### Comment · Reviewer_nisC · 2024-06-03
> >
> > The authors have provided detailed responses to address my questions. I would like to maintain my positive rating of this paper.

---

### Official Review · Reviewer_rGMR · 2024-05-11

**Rating:** 6
**Confidence:** 3
**Ethics Flag:** 1

**Summary:**

The authors propose a probing framework based on Structural Causal Models (SCMs) in order to uncover the latent variables LMs learn. As their main contributions, the authors argue that they show that LMs learn the "underlying semantics of the language" (page 2), and they can learn to generalize to unseen data (page 2).

While I agree with the authors that there is much to be explored when it comes to developing robust and reliable probing mechanisms and I appreciate the presentation of SCMs, I fail to see how this work contributes on top of already existing work and what its main audience is.

[edit of the review during discussion period: I adjusted my overall score from 4 to 6.]

**Questions To Authors:**

Very minor note: There are multiple cases where the text says "casual" where it's supposed to say "causal" (I assume).

The authors might find the following paper interesting to make connections to given the parallels to SCMs, a focus on the training data, the navigational instruction task, and connections to generalizability and data augmentations (Inducing Causal Structure for Interpretable Neural Networks; Geiger, Wu, Lu et al.; ICML 2022).

**Reasons To Accept:**

The overall topic is relevant to the NLP community. The paper is clearly structured and SCMs as well as probing mechanisms are intuitively introduced.

**Reasons To Reject:**

My main concern is that the overall goal of the paper is unclear to me.

If the paper is about investigating whether LMs learn "the underlying semantics of the language" (page 2), then this is very well-established already. It is a direct consequence of the good performance of LMs on relational similarity prediction (king-man+woman=queen; see e.g., Vylomova, Rimell, Cohn, Baldwin 2016), using semantic representations for semantic cluster analyses, and vision-language representation space alignment work. If the paper aims to establish overall that LMs latently encode a variety of semantic variables (which I'm again not sure adds much to existing work), this should be evaluated in a suite of experiments. It is my intuition that this isn't the main goal. However, especially the framing of the results, and also the narrative in the abstract and introduction appear to put a significant focus here instead of using this task as a "proof-of-concept".

If the paper is intended as an argument for using this particular SCM-based framework as a probing mechanism, the paper needs to focus on establishing it against alternative probing methods, elaborating on how to use it for a variety of potential downstream tasks, as well as establishing its effectiveness. For example, in the results, the paper solely focuses on the implications for the model under investigation but under this framing, I would instead expect a focus on establishing the method.

Lastly, I'm confused about the way in which the authors present this work as "interven[ing] on the model's training data" [page 9], as opposed to other work that intervenes on model representations. However, intervening on training data to understand the "causal effect of dataset statistics" [page 9] is the general idea of traditional data augmentation and model generalizability setups (e.g., Perez and Wang 2017; Liu, Kusner, and Blunsom 2021). How does this setup situate within the established data augmentation and generalizability frameworks?

---

> ### Author Rebuttal · Authors · 2024-05-31
>
> We thank the reviewer for taking the time to engage with our submission! If there are any remaining concerns we can address, please do let us know in the discussion period.
>
> > If the paper is about investigating whether LMs learn "the underlying semantics of the language" ...
>
> Our paper establishes a framework for studying the question of whether **LMs are capable of inducing the causal structure which is latent in their training data**. While in the setting of our experiments, the causal structure happens to come from the formal semantics of the programming language, in general, causal structure and semantics are distinct concepts.
>
> To the best of our knowledge, **our results are the first to show that LMs can infer free latent variables from text**, which suggests that LMs have the ability to perform causal induction. Note that none of the examples in the review relate to an LM’s ability to perform causal induction.
>
> > If the paper is intended as an argument for using this particular SCM-based framework.. against alternative probing methods
>
> We summarize some major implications of our framework for probing in the remarks titled “Interventions, and probing for non-causal latent variables” on page 7. Briefly, Proposition 1 presents precise, formal conditions which guarantee the probe is properly controlled. The converse is also true: if the conditions laid out by Proposition 1 are NOT satisfied, then the results *could* be measuring either 1) the ability of the probe to learn the task (confounding) OR 2) the “naturalness” of the original task (bias). Indeed, existing probing frameworks produce baselines that are biased (“Frameworks for probing” in the Related works on page 9).
>
> We will commit to dedicating a portion of the extra content page to a more detailed discussion that situates our contributions against other probing methodologies if the paper is accepted.
>
> > How does this setup situate within the established data augmentation and generalizability frameworks?
>
> At a high level, the works cited in the review are generally concerned with “(Why / how) does including interventions (data augmentation) in the training data improve generalization?”
>
> In contrast, interventions are simply a technique we employ to answer the question “Can LMs learn latent causal variables?”  A model is only ever trained on either the original OR the intervened data, NEVER both; and in fact a positive result requires the model to perform poorly on the intervened data.

---

> > ### Comment · Reviewer_rGMR · 2024-06-01
> >
> > I thank the authors for engaging with my review. Upon revisiting the paper after reading the author response, my questions were largely resolved and I changed my overall rating accordingly. I can see now how this work comes together considering this overall framing. The only remaining concerns are with the limited presentation of the results in the paper and that I believe the paper would benefit from the extra discussion the authors indicate they intend to include.

---

> > > ### Author Response · Authors · 2024-06-04
> > >
> > > Thank you for the update!
> > >
> > > > The only remaining concerns are with the limited presentation of the results in the paper and that I believe the paper would benefit from the extra discussion the authors indicate they intend to include.
> > >
> > > We agree these suggestions make for a stronger paper and will be sure to do this.

---

### Official Review · Reviewer_Jppb · 2024-05-14

**Rating:** 7
**Confidence:** 4
**Ethics Flag:** 1

**Summary:**

The paper proposes integrating Structural Causal Models (SCM) with the existing probing techniques in NLP. This method aims to refine how we interpret probing results, providing insights into the information that language models (LMs) encode from their training data.

One key contribution of this paper is its focus on the probe's capacity, addressing a common shortcoming in traditional probing studies. By using causal mediation analysis, the paper effectively distinguishes the information genuinely encoded by the LM from what is learned independently by the probe. This approach is crucial as it aims to ensure that high probing accuracy reflects the LM's capabilities rather than the probe's.

The paper also advances beyond merely associating high probing accuracy with an LM's ability to understand causal structures. Instead, it uses causal models to rigorously test hypotheses about the causal information that LMs might capture, providing a more nuanced interpretation of probing results.

However, the reliance on SCMs assumes clear causal relationships within the training data, which may not always be evident, especially in natural language settings where such relationships can be complex and less defined. This assumption is a significant limitation and could restrict the method's application to more naturalistic datasets where causal relationships are not explicitly outlined.

The paper demonstrates its approach in controlled experimental setups. However, extending this framework to broader, real-world datasets and demonstrating its effectiveness in these contexts could further validate its practicality and impact.

**Reasons To Accept:**

1. **Methods**: Integrating Structural Causal Models (SCM) with language model probing is both novel and interesting. This methodology introduces a new way to interpret what language models learn from data, focusing on understanding causal relationships rather than accuracy metrics.

2. **Tackling limitations of probing**: The paper tackles the limitations of current probing practices by focusing on the probe's capacity to learn independently of the language model. This is important for obtaining more accurate interpretations of model behaviors and capabilities, addressing a gap in how probes are currently used.

**Reasons To Reject:**

1. **Assumptions on causal relationships**: The method assumes clear causal relationships within the training data for language models. In practice, especially with complex real-world data, these relationships are often unclear or confounded by other factors.

2. **Limited experiments**: The paper primarily validates its approach in controlled settings. To strengthen the case for publication, it would need to demonstrate effectiveness with real-world datasets, which are generally less structured.

3. **Overfitting**: There is a concern that the causal models could be overfitted to specific characteristics of the training data, given the high dimensionality of language model embeddings. This risk could lead to incorrect conclusions about what the language models are learning.

---

> ### Author Rebuttal · Authors · 2024-05-31
>
> We are glad the reviewer appreciates our contributions to the state of probing! We hope our rebuttal can address the remaining concerns
>
> > Assumptions on causal relationships
>
> While we agree this makes it difficult to apply the technique directly on real world data, we do want to provide another perspective, which is that our assumptions put the risks of probing on real world data in a *mathematically precise language*. Specifically, the lack of a clear causal relationship means that we cannot hope for a valid baseline in the exact sense of Proposition 1, so extra care should be taken when interpreting results against a baseline measurement.
>
> > Limited experiments
>
> We chose to test our methodology in controlled settings so that we could follow the methodology as faithfully as possible. We felt that, given that the methodology is itself a major contribution of the paper, conducting our experiments under “ideal” circumstances would allow us to demonstrate the benefits of the new framework as cleanly as possible and minimize concerns around whether the results could be due to a misapplication of the methodology.
>
> We would also like to emphasize that our experiments are, to the best of our knowledge, the first to show that LMs *can* infer free latent variables from text. We think there’s an argument for conducting experiments in a controlled environment where the complexity of real-world data does not intrude on experimental design.
>
> > Overfitting
>
> Our understanding of this point is that the high dimensionality of the LM representations may allow the probe to “find something which is not really there”. However, having a valid baseline (Proposition 1) actually controls for this already: loosely speaking, a valid baseline would benefit equally from the high dimensionality of the LM representations.
>
> If this is not the correct reading, please let us know and we would be happy to respond in the discussion!

---

> > ### Comment · Reviewer_Jppb · 2024-06-04
> >
> > Thank you for addressing my concerns.
> >
> > While I still would have liked to see additional experiments in more natural settings, this paper is insightful in its current state and I support accepting it.

---

### Decision · Program_Chairs · 2024-07-10

**Decision:**

Accept

**Comment:**

The paper revisits an old chestnut in the NLP community -- probing. What excited me most about the paper was the application of SCMs. All reviewers highlighted the causal framework the paper introduces as a strength. The only reasons to reject given by the reviewers seem to focus on the number of experiments (after investigating the experiments myself, I disagree), and a limited number of latent variables. I think, however, that the importance of the problem trumps those concerns. Understanding and getting into the causal mechanisms that underlie language models is an important endeavor.